# *ACADM* Frameshift Variant in Cavalier King Charles Spaniels with Medium-Chain Acyl-CoA Dehydrogenase Deficiency

**DOI:** 10.3390/genes13101847

**Published:** 2022-10-13

**Authors:** Matthias Christen, Jos Bongers, Déborah Mathis, Vidhya Jagannathan, Rodrigo Gutierrez Quintana, Tosso Leeb

**Affiliations:** 1Vetsuisse Faculty, Institute of Genetics, University of Bern, 3001 Bern, Switzerland; 2Neurology and Neurosurgery Service, The School of Veterinary Medicine, College of Medical, Veterinary and Life Sciences, University of Glasgow, Glasgow G12 8QQ, UK; 3University Institute of Clinical Chemistry, Inselspital, Bern University Hospital, University of Bern, 3010 Bern, Switzerland

**Keywords:** *Canis lupus familiaris*, dog, neurology, metabolism, fatty acid disorder, seizure, precision medicine

## Abstract

A 3-year-old, male neutered Cavalier King Charles Spaniel (CKCS) presented with complex focal seizures and prolonged lethargy. The aim of the study was to investigate the clinical signs, metabolic changes and underlying genetic defect. Blood and urine organic acid analysis revealed increased medium-chain fatty acids and together with the clinical findings suggested a diagnosis of medium-chain acyl-CoA dehydrogenase (MCAD) deficiency. We sequenced the genome of the affected dog and compared the data to 923 control genomes of different dog breeds. The *ACADM* gene encoding MCAD was considered the top functional candidate gene. The genetic analysis revealed a single homozygous private protein-changing variant in *ACADM* in the affected dog. This variant, XM_038541645.1:c.444_445delinsGTTAATTCTCAATATTGTCTAAGAATTATG, introduces a premature stop codon and is predicted to result in truncation of ~63% of the wild type MCAD open reading frame, XP_038397573.1:p.(Thr150Ilefs*6). Targeted genotyping of the variant in 162 additional CKCS revealed a variant allele frequency of 23.5% and twelve additional homozygous mutant dogs. The acylcarnitine C8/C12 ratio was elevated ~43.3 fold in homozygous mutant dogs as compared to homozygous wild type dogs. Based on available clinical and biochemical data together with current knowledge in humans, we propose the *ACADM* frameshift variant as causative variant for the MCAD deficiency with likely contribution to the neurological phenotype in the index case. Testing the CKCS breeding population for the identified *ACADM* variant is recommended to prevent the unintentional breeding of dogs with MCAD deficiency. Further prospective studies are warranted to assess the clinical consequences of this enzyme defect.

## 1. Introduction

Medium-chain fatty acids (MCFAs) are monocarboxylic acids with a hydrocarbon chain of six to twelve carbon atoms in length (C6–C12) [1]. They either are taken up in the gastrointestinal tract, or are derived through β-oxidation of long-chain fatty acids, catalyzed by the enzyme very long chain acyl CoA dehydrogenase [2,3]. Through further β-oxidation, now mediated by medium-chain acyl-CoA dehydrogenase (MCAD), MCFAs serve as energy source for the body [4].

In humans, mitochondrial fatty acid β-oxidation disorders are a heterogeneous group of inherited diseases with a wide range of clinical presentation [1]. MCAD deficiency is the most frequently diagnosed disease in this group [5], for which widespread screening in European newborns has shown that approximately 1/8000–1/20,000 are affected by MCAD deficiency [6,7,8].

In MCAD deficient patients, unmetabolized MCFAs accumulate in different tissues [9,10]. As a result of the impaired β-oxidation, affected people are not able to produce sufficient energy out of ketone bodies during times of extended fasting or acute stress [5]. They present to the emergency room with an acute crisis of hypoketotic hypoglycemia. Clinically, such a crisis manifests as ‘Reye-like symptoms’, which consist of vomiting, lethargy, hepatomegaly and liver dysfunction that may eventually result in encephalopathy, seizures and even coma and death [11].

MCAD deficiency is caused by variants in the *ACADM* gene (OMIM #201450) [12]. Many distinct disease-causing variants have been identified in different human populations [8,13]. A targeted mouse model for MCAD deficiency has been developed [14], but otherwise the disease has rarely been observed in animals. In a single Cavalier King Charles Spaniel (CKCS), MCAD deficiency was suspected based on the results of blood and urine organic acid levels [15], but the causative genetic variant was not investigated in this dog.

This study was initiated after the presentation of a CKCS with a history of complex focal seizures and laboratory findings strongly resembling human MCAD deficiency. The goal of the study was to characterize the clinical and metabolic phenotype and to investigate a possible underlying causative genetic defect.

## 2. Materials and Methods

### 2.1. Clinical Examination and Investigations

A single, 3-year-old, male neutered CKCS dog originating in the United Kingdom was investigated. Both parents were reportedly healthy, the health status of siblings was unknown. The dog was presented to the Small Animal Hospital of the University of Glasgow for investigations of suspected focal seizures. Blood was taken for hematology and serum biochemistry. Magnetic resonance imaging (MRI) of the brain was performed with a 1.5 Tesla machine (1.5T Magnetom, Siemens, Erlangen, Germany and included T2-weighted sagittal, dorsal and transverse views and the following transverse view: fluid attenuated inversion recovery (FLAIR), Gradient echo (t2*), T1-weighted pre- and post-contrast sequences (gadopentate dimeglumine; Magnevist, Bayer Schering Pharma AG, Berlin, Germany). A cerebrospinal fluid sample was taken for total and differential cell counts, and protein levels. Finally, urine was submitted for organic acid analysis and blood for acylcarnitine levels to an external human laboratory. A control sample of a clinically healthy dog was sent to compare the acylcarnitine levels, as there are no published reference ranges for dogs.

### 2.2. DNA Extraction

Genomic DNA was isolated from EDTA blood with the Maxwell RSC Whole Blood Kit using a Maxwell RSC instrument (Promega, Dübendorf, Switzerland). In addition to the affected dog, 162 blood samples from CKCS, which had been donated to the Vetsuisse Biobank, were used. Most of these additional samples were obtained during an MRI screening program for syringomyelia in the Swiss and German CKCS population. Potential MCAD deficiency had not been investigated in these dogs.

### 2.3. Whole-Genome Sequencing

An Illumina TruSeq PCR-free DNA library with ~413 bp insert size of the affected dog was prepared. We collected 280 million 2 × 150 bp paired-end reads corresponding to 30.9 × coverage on a NovaSeq 6000 instrument (Illumina, San Diego, CA, USA). Mapping to the UU_Cfam_GSD_1.0 reference genome assembly was performed as described [16]. The sequence data were deposited under the study accession PRJEB16012 and the sample accession SAMEA10644719 at the European Nucleotide Archive. Genome sequence data of 923 control dogs of diverse breeds were also included in the analysis (Appendix A).

### 2.4. Variant Calling

Variant calling was performed using GATK HaplotypeCaller [17] in gVCF mode as described [16]. To predict the functional effects of the called variants, SnpEff v 5.0e software [18], together with UU_Cfam_GSD_1.0 reference genome assembly and NCBI annotation release 106, was used.

### 2.5. Gene Analysis

Numbering within the canine *ACADM* gene corresponds to the NCBI RefSeq accession numbers XM_038541645.1 (mRNA) and XP_038397573.1 (protein).

### 2.6. Allele Specific PCR and Sanger Sequencing

Primers 5′-GAG TAA AGG CCA GTT CTT TGG A-3′ (Primer F) and 5′-CCT GGT AAC CCA GAA ACA TCA-3′ (Primer R) were used for the generation of an amplicon containing the *ACADM*:c.444_445delinsGTTAATTCTCAATATTGTCTAAGAATTATG variant. PCR products were amplified from genomic DNA using AmpliTaq Gold 360 Master Mix (Thermo Fisher Scientific, Reinach, Switzerland). Product sizes were analyzed on a 5200 Fragment Analyzer (Agilent, Basel, Switzerland). Direct Sanger sequencing of the PCR amplicons on an ABI 3730 DNA Analyzer (Thermo Fisher Scientific, Reinach, Switzerland) was performed after treatment with exonuclease I and alkaline phosphatase. Sanger sequences were analyzed using the Sequencher 5.1 software (Gene Codes, Ann Arbor, MI, USA).

### 2.7. Acylcarnitine Screening

Five animals (3 females and 2 males) were included in each genotype group (homozygous wildtype, heterozygous, homozygous mutant). The initially investigated clinical case was included in the homozygous variant group. All animals were adult (between 2.7 and 9.9 years of age). Whole blood samples of the animals were stored at −20 °C until analysis. Acylcarnitines were analyzed with modification of a previously published protocol [19]. In short, 20 μL of hemolyzed whole blood, 20 µL acetonitrile (ACN), and 360 µL of water containing deuterated acylcarnitines internal standards were pipetted into an Eppendorf tube. The tubes were vortexed, set 5 min into an ultrasound bath and centrifuged at 12,000× *g*. Next, 20 µL of the clear supernatant was transferred into a HPLC vial containing 180 µL of ACN. Then, 2 µL was injected into the liquid chromatography mass spectrometer (HPLC-MS/MS, Waters Xevo TQ-S with Acquity I-Class 2D UPLC). The HPLC was mounted with an ACQUITY UPLC BEH Amide column (2.1 × 100 mm, 1.8 µm, Waters), the eluents were A (water:ACN (1:1) 10 mM ammonium formate with 0.15% formic acid) and B (water:ACN (5:95) 10 mM ammonium formate with 0.15% formic acid acetonitrile) at a flow rate of 0.4 mL/min. The gradient was: 100% B until 1.5 min, 74% B at 6 min, 22% B at 8 until 10 min then back to 100% B. Total run time was 15 min. The acylcarnitines were analyzed with positive electrospray ionization using multiple reaction monitoring (MRM) ion scan mode. Absolute quantification was achieved with a 6-point calibration curve.

## 3. Results

### 3.1. Clinical History, Examination and Investigations

A male neutered CKCS, born out of reportedly healthy parents, was presented at the age of 1.5 years, with an acute history of suspected complex focal seizures including prolonged lethargy, being less responsive and proprioceptive ataxia. These episodes initially occurred several times a week, lasting from 20 min to multiple hours during which the dog was mainly lethargic. General physical examination and neurological examination were normal. Complete blood count and serum biochemistry profile were within normal limits. MRI imaging of the brain revealed breed-related changes including occipital malformation with mild cerebellar herniation, medullary kinking and syringohydromyelia, consistent with canine Chiari-like malformation and syringomyelia (CMSM). No other abnormalities of the brain were detected. The results of the cerebrospinal fluid (CSF) analysis collected at the cerebellomedullary cistern showed mild albuminocytological dissociation (total nucleated cell count: 0 cells/µL, RI < 5 cells/µL; protein concentration 40 mg/dL, RI < 25 mg/dL). The dog was prescribed 40 mg/kg levetiracetam three times a day, however this resulted in severe sedation. The levetiracetam dose was therefore lowered to 25 mg/kg three times a day and 3 m/kg phenobarbital twice a day was started, which resulted in a partial response as the seizures decreased in frequency and intensity. The patient remained stable for 3 months before the seizure interval increased again and particularly the lethargy remained present up to 24 h. The dog would return to normal the following morning. An increase in the phenobarbital dose was not accompanied by an improvement. Given the unusual presentation, urine was analyzed for organic acids and revealed significant excretion of hexanoylglycine and a peak of suberic acid, highly suggestive of a fatty acid β-oxidation disorder. A follow-up test consisted of blood spot acylcarnitine analysis and revealed an increase in C6, C8 and C10:1 acylcarnitines, as judged against human adult reference intervals and a clinically normal dog (Appendix A). Extrapolating from human patients, and in comparison with the control dog, the acylcarnitine profile was consistent with a diagnosis of medium-chain acyl-CoA dehydrogenase deficiency.

In addition to 25 mg/kg levetiracetam three times a day and 3.75 mg/kg phenobarbital twice a day, the dog was prescribed a low-fat diet and a midnight snack consisting of carbohydrates. Prolonged periods of fasting and formulas that contained medium-chain triglycerides as primary source of fat were also advised to avoid. This management protocol correlated with a complete resolution of clinical signs for the following 6 months. The anticonvulsant medication was therefore reduced to subtherapeutic levels. However, this was reversed as the dose reduction resulted in an increase in seizure frequency. The blood spot acylcarnitines were repeated to test sufficient free carnitine levels and these were found within normal limits (Appendix A). At the time of writing, the dog has been stable for 9 months on 25 mg/kg levetiracetam three times a day, 3 mg/kg phenobarbital twice a day and a low-fat diet, with no further major seizures and a repeated normal neurological examination.

### 3.2. Genetic Analysis

As clinical and laboratory findings resembled human patients and a previously published CKCS with suspected MCAD deficiency [15], we hypothesized that the phenotype in the affected dog was due to a variant in the *ACADM* gene. Hence, *ACADM* was investigated as the top functional candidate gene. We sequenced the genome of the affected dog and searched for private homozygous variants that were not present in the genome sequences of 923 control dogs of diverse breeds (Table 1 and Appendix A).

The automated analysis identified three closely spaced homozygous private protein-changing variants in *ACADM*. Visual inspection of the short read alignments in the region revealed that these three initially separately called variants actually represented just one single insertion-deletion variant. This variant, XM_038541645.1:c.444_445delinsGTTAATTCTCAATATTGTCTAAGAATTATG, leads to a frameshift and is predicted to truncate 267 codons or roughly 63% of the wild type MCAD open reading frame, XP_038397573.1:p.(Thr150Ilefs*6). On the genomic level, the variant can be designated as Chr6:71,401,388_71,401,389delinsCATAATTCTTAGACAATATTGAGAATTAAC (Figure 1).

We genotyped the variant in a cohort of 162 CKCS that were not closely related to the index case and sampled during an independent study (Table 2). This experiment revealed 52 heterozygous carriers and 12 homozygous mutant dogs. The genotype distribution did not significantly deviate from Hardy–Weinberg equilibrium. The frequency of the putative disease allele was 23.5% in the investigated CKCS population.

### 3.3. Acylcarnitine Measurements

To confirm the functional impact of the *ACADM* variant on fatty acid metabolism, acylcarnitines were measured in five dogs of each genotype (Appendix A). Biomarkers of MCAD deficiency, C8- and C10:1-carnitines were elevated in all homozygous dogs compared to the five WT dogs. The specific C8/C10 and C8/C12 ratios used for diagnosing MCAD deficiency in humans were elevated 1.3 and 2.9-fold in heterozygous dogs, respectively and 11 and 65-fold in homozygous variant dogs, respectively as compared to wild type dogs (P_ANOVA_ C8/C10= 5.1 × 10^−5^; P_ANOVA_ C8/C12= 1.4 × 10^−5^, Figure 2).

## 4. Discussion

In this study, we identified a homozygous *ACADM* frameshift variant in a CKCS with a history of complex focal seizures including lethargy and highly elevated MCFA metabolites in blood and urine metabolic testing. The clinical phenotype of the affected CKCS resembled human patients with MCAD deficiency and variants in the human *ACADM* gene (OMIM #201450) [12]. The investigated dog also showed striking clinical and biochemical similarities to a previously described CKCS with aciduria and elevated levels of urine hexanoylglycine and plasma acylcarnitines [15]. The plasma acylcarnitine C8/C12 ratio in the previously investigated case was at 28, which is comparable to the ratios found in the homozygous mutant dogs of our study (range 20–52, median 28). Typical pathological C8/C10 and C8/C12 ratios in human newborns range between 1.6–18 and 4.4–449, respectively [20]. In humans, the acylcarnitine biomarkers and ratios remain elevated even between decompensation episodes and under appropriate treatment (carnitine supplementation; avoiding prolonged fasting and lipolysis) [21].

Somewhat unexpectedly, the mutant allele was quite common in a representative population of CKCS that were examined for the presence of syringomyelia in a Swiss/German screening program. The CKCS breed is genetically predisposed for the occurrence of Chiari-like malformation and syringomyelia, which may result in phantom scratching, pain, and neurological deficits such as scoliosis, weakness and proprioceptive impairment [22]. An association between Chiari-like malformation and epileptic seizures was hypothesized [23], but could not be confirmed in an experimental investigation [24]. The identified *ACADM* variant now provides a compelling new candidate variant, which might be responsible for a part of the seizure phenotypes that are observed in the CKCS breed. Clinical signs due to Chiari-like malformation and/or syringomyelia and MCAD deficiency are partially overlapping and may be very difficult to disentangle in a clinical setting. The most objective way of differentiating epileptiform seizures would be by recording the electrical activity of the brain using electroencephalography, but this is technically impractical for several reasons in veterinary settings [25]. Further prospective studies are needed to better differentiate between those diseases in CKCS and to evaluate the clinical impact of the observed enzyme deficiency in some dogs of this breed.

MCAD deficiency in dogs seemingly does not clinically manifest as severe as in humans. However, our data show a clear increase in MCFAs in *ACADM* homozygous mutant dogs. This might point to an additional compensatory mechanism in the dog, which prevents or dampens the manifestation of clinical consequences of elevated MCFAs. In humans, phenotypic diversity ranging from sudden neonatal death to asymptomatic status has previously been reported. Human patients with complete loss of MCAD activity can also remain asymptomatic, suggesting that additional genetic or environmental factors may play a role in the phenotypic diversity [26,27,28].

Additional genetic or environmental factors are also likely to modulate the phenotype in MCAD deficient dogs. The improvement of clinical signs upon changing to a low-fat diet in our index case indicates that the diet has a major influence on the clinical phenotype. At this time, we cannot exclude the possibility that additional genetic factors also modified the clinical phenotype. While our data conclusively demonstrate that the *ACADM* frameshift variant causes MCAD deficiency and the biochemical alterations in the lipid metabolism, it is not yet fully clear whether the MCAD deficiency alone is responsible for the clinical phenotype or whether additional environmental and/or genetic risk factors are required for the expression of clinical signs. The identification of the *ACADM* frameshift variant enables genetic testing for MCAD deficiency and will facilitate future prospective studies to clarify this important question.

In humans, newborn screening programs are now well established, but prior to this, the majority of human MCAD deficiency cases presented at young age (before 2 years) [11]. No newborn screenings are performed in dogs, and it is currently unknown if MCAD deficiency could have an impact in CKCS neonatal mortality. Previous studies have reported a high percentage of perinatal mortality in CKCS as is the case for many purebred dogs [29,30]. We did not observe a significant deviation from Hardy–Weinberg equilibrium in our cohort of 162 dogs. Hence, a possible influence of the *ACADM* variant on neonatal mortality in the breed is presumably low. Nonetheless, further prospective studies might be considered to investigate if MCAD deficiency plays a role in CKCS neonatal mortality.

## 5. Conclusions

We identified a dog with MCAD deficiency that clinically, biochemically and genetically resembled human patients with variants in the *ACADM* gene. The putative disease allele was common in a representative CKCS cohort and might contribute to seizure phenotypes that are observed in the breed. Our data enable genetic testing to establish a diagnosis in dogs with suspected MCAD deficiency and to prevent the unintentional breeding of further dogs with MCAD deficiency. Further prospective studies are needed to assess the clinical consequences of MCAD deficiency in the CKCS breed.

## Figures and Tables

**Figure 1 genes-13-01847-f001:**
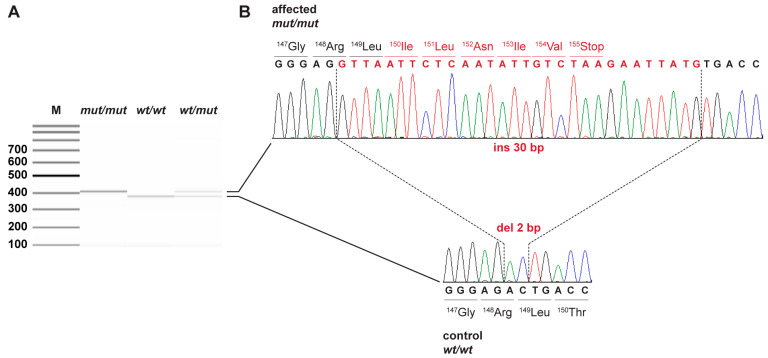
Details of the *ACADM*: c.444_445delinsGTTAATTCTCAATATTGTCTAAGAATTATG variant. (**A**) Fragment Analyzer bands of PCR products from samples of all three genotypes show the expected 28 bp difference in length of the wild type and mutant products. (**B**) Sanger sequencing electropherograms of the dog affected by MCAD deficiency (top) and a control dog (bottom) illustrate the deletion of 2 bp with simultaneous insertion of 30 bp in exon 6 of the *ACADM* gene. Altered nucleotide and amino acid sequences are indicated in red.

**Figure 2 genes-13-01847-f002:**
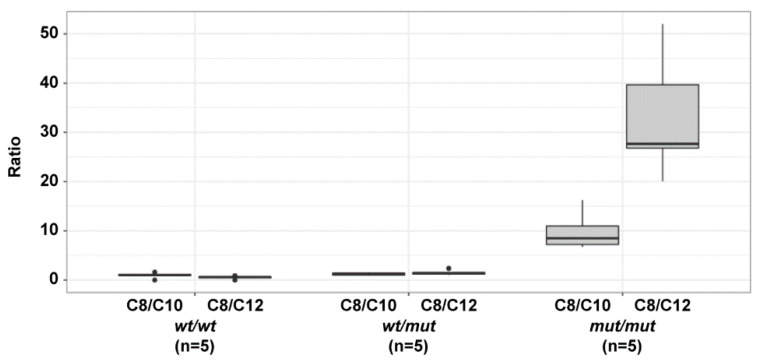
Acylcarnitine C8/C10 and C8/C12 ratios in the blood of dogs with different *ACADM* genotypes.

**Table 1 genes-13-01847-t001:** Results of variant filtering in the affected dog against 923 control genomes.

Filtering Step	Homozygous Variants
All variants in the affected dog	3,063,158
Private variants	1562
Protein-changing private variants	10
Private protein changing variants in *ACADM* candidate gene	3

**Table 2 genes-13-01847-t002:** Genotype distribution at the *ACADM* frameshift variant in 162 CKCS.

Genotype Frequency	*wt/wt*	*wt/mut*	*mut/mut*
Number (Percentage) of dogs	98 (60.5%)	52 (32.1%)	12 (7.4%)

## Data Availability

The accessions for the sequence data reported in this study are listed in Appendix A.

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
