# Peer review of "ACADM Frameshift Variant in Cavalier King Charles Spaniels with Medium-Chain Acyl-CoA Dehydrogenase Deficiency"

_genes, 2022, doi:10.3390/genes13101847_

Round 1
Reviewer 1 Report
The study reports the identification of a variant in the ACADM gene causative of a fatty acid beta-oxidation disorder in Cavalier King Charles Spaniel dogs. The genomic study is consistent with current analytical approaches. Overall, the delins variant identification is well designed and conducted.
Some comments to be addressed on the validation at the population level
1. Line 82-87: please specify if pedigree data were available and if it has been possible to check whether at least the majority of dogs were unrelated
2. Line 176: “a previously published dog with suspected MCAD deficiency, etc”, please add the reference.
3. Line 202: The validation analysis at the population level provided the variant allele frequency on a population of 162 CKCS. The results of this screening have been discussed in the manuscript; however, I please ask the authors to address the following comment.
A variant allele frequency of 23.5% is a high percentage for a rare disease mutation, even autosomal recessive, as you also noted (line 229). It is not clear to me the MCAD clinical status of those dogs, collected for another disease screening not proved to be related to MCAD clinical disease (Line 235 ref 24). The authors state MCAD status data were not available. Thus, it was not possible to state if those controls were clinically MCAD healthy, but it is possible to assume that the 12 mut/mut dogs (7,4%, Table 2) were not clearly affected by the disease. In conclusion, all the samples analyzed in the whole study (including the four out of five homozygous mutants of the acylcarnitine screening -Line 116) were not clearly clinically affected. The only affected was the dog from the UK, being your controls collected in Switzerland and Germany (but not in UK). The effect of the delins is evident on the acylcarnitine ratios, but a discrepancy occurs between the genotype and the macro clinical signs. The authors discuss this discrepancy concluding that the variant might be responsible for “a part of the seizure phenotypes that are observed in the CKCS breed” and stating that “further studies are warranted to assess the clinical consequences…”.
In general, and in perspective, it is important to distinguish those variations that can be important to make help vet clinical analysis and therapy at the individual (or perhaps line) level, from those variations that are important at a general breed level and can be widely suggested to the breeders for selection.
In the present manuscript, the general “recommendation” of the genetic test screening in the CKCS breeding population for the identified ACADM variant to prevent the unintentional breeding of dogs with MCAD deficiency (such as in Lines 31-32; 267-268), thus for selection purposes, should be avoided, being the effect on the clinical disease still not clarified.
Author Response
(1)
Line 82-87: please specify if pedigree data were available and if it has been possible to check whether at least the majority of dogs were unrelated.
Response: We have pedigree data on most of these 162 dogs. This is admittedly a "convenience cohort" with many related dogs. However, as it represents a large proportion of the Swiss CKCS population, we nonetheless think that the reported allele frequency is at least representative for the Swiss CKCS population. We did not change the manuscript with respect to the comment of the reviewer as the pedigree relationships between these dogs cannot easily be described in a short verbose text.
(2)
Line 176: “a previously published dog with suspected MCAD deficiency, etc”, please add the reference.
Response: We added the reference as suggested and changed "dog" into "CKCS" as we consider it important to note that the previously published case was from the same breed as the dog in our study.
(3)
Line 202: The validation analysis at the population level provided the variant allele frequency on a population of 162 CKCS. The results of this screening have been discussed in the manuscript; however, I please ask the authors to address the following comment.
A variant allele frequency of 23.5% is a high percentage for a rare disease mutation, even autosomal recessive, as you also noted (line 229). It is not clear to me the MCAD clinical status of those dogs, collected for another disease screening not proved to be related to MCAD clinical disease (Line 235 ref 24). The authors state MCAD status data were not available. Thus, it was not possible to state if those controls were clinically MCAD healthy, but it is possible to assume that the 12 mut/mut dogs (7,4%, Table 2) were not clearly affected by the disease. In conclusion, all the samples analyzed in the whole study (including the four out of five homozygous mutants of the acylcarnitine screening -Line 116) were not clearly clinically affected. The only affected was the dog from the UK, being your controls collected in Switzerland and Germany (but not in UK). The effect of the delins is evident on the acylcarnitine ratios, but a discrepancy occurs between the genotype and the macro clinical signs. The authors discuss this discrepancy concluding that the variant might be responsible for “a part of the seizure phenotypes that are observed in the CKCS breed” and stating that “further studies are warranted to assess the clinical consequences…”.
In general, and in perspective, it is important to distinguish those variations that can be important to make help vet clinical analysis and therapy at the individual (or perhaps line) level, from those variations that are important at a general breed level and can be widely suggested to the breeders for selection.
In the present manuscript, the general “recommendation” of the genetic test screening in the CKCS breeding population for the identified ACADM variant to prevent the unintentional breeding of dogs with MCAD deficiency (such as in Lines 31-32; 267-268), thus for selection purposes, should be avoided, being the effect on the clinical disease still not clarified.
Response: We thank the reviewer for this important comment. We agree with the reviewer that it is unlikely that all CKCS with the homozygous mutant genotype will develop clinically overt signs (= incomplete penetrance). Incomplete penetrance regarding the clinical signs is even documented in human patients, although they tend to exhibit a more severe clinical phenotype than dogs in general.
In the cohort of 162 dogs, at least one of the 12 homozygous mutant dogs had a clinical history including seizures. Unfortunately, with the limited data available, we could not reliably establish whether the seizures in this dog might have been due to its MCAD deficiency.
We respectfully disagree with the opinion of the reviewer that it is premature to recommend genetic testing in the breeding program. We think that we have conclusively shown that homozygous mutant dogs have a defect in their lipid metabolism and may develop a relatively severe disease. At the same time, the mutant allele is relatively common in the population. A sensible breeding program that does not exclude any dog from breeding based on its ACADM genotype, but rather requires that carriers (and possibly homozygous mutant dogs) may only be mated to clear dogs will ensure that no further MCAD deficient puppies are born, while at the same time preserving the remaining genetic diversity in the breed. CKCS are infamous for having a large number of common breed-specific morbidities. To the best of our knowledge, in Norway, the breed may soon be banned due to its multiple health issues. While we concede that MCAD deficiency may not represent the most severe health problem in CKCS, it can now easily be addressed without any negative consequences for all the other health issues. We consider it therefore ethically justified (and necessary) to recommend the introduction of genetic testing for breeding purposes.
We agree with the reviewer that the genetic test will also be a valuable diagnostic tool for veterinarians to rule out or confirm a suspected MCAD deficiency in CKCS. We revised the abstract and conclusions accordingly.
Reviewer 2 Report
It is a well-written and interesting manuscript describing a new genetic disease in dogs, which counterpart is well known in humans. The authors used appropriate molecular and bioinformatic methods to identify a putative causative mutation.
Remarks:
1. Material. It would be interesting and important, if possible, to add information concerning health status and genotypes of the affected dog’s relatives (full sibs, parents, other relatives).
2. Results.
a) information concerning health status of five dogs homozygous for the mutation, which were analyzed for the impact of the ACADM variant on fatty acid metabolism should be added (lines 205-206), should be added. By the way the results are presented in Supp. Table 2, not in Suppl. Table 3,
a) some information concerning biological function of genes (apart of the ACADM), in which private variants were identified (10 variants mentioned in Table 1) should be added (e.g. in a supplementary table).
3. Discussion. It would be valuable to discuss mechanisms, which can be responsible for variation of the disease severity in the affected dogs (incomplete penetrance or incomplete expressivity of the mutation, oligogenic inheritance, epigenetic mechanisms, etc?).
Author Response
(1)
Material. It would be interesting and important, if possible, to add information concerning health status and genotypes of the affected dog’s relatives (full sibs, parents, other relatives).
Response: Thank you very much for alerting us to this important omission. The parents were healthy, information on other relatives is unfortunately not available. We added this information to chapters 2.1 (Material and Methods) and 3.1 (Results).
(2)
Results:
- a) information concerning health status of five dogs homozygous for the mutation, which were analyzed for the impact of the ACADM variant on fatty acid metabolism should be added (lines 205-206), should be added. By the way the results are presented in Supp. Table 2, not in Suppl. Table 3,
Response: Unfortunately, we do not have consistent health data on the genotyped dogs. We used archived EDTA blood samples for genotyping and the acylcarnitine measurements. The health status of the sampled dogs is unknown.
Thank you for spotting the incorrect reference to the supplementary data. This error has been revised.
- b) some information concerning biological function of genes (apart of the ACADM), in which private variants were identified (10 variants mentioned in Table 1) should be added (e.g. in a supplementary table).
Response: We added the requested information (OMIM numbers and OMIM phenotypes) to Table S3.
(3)
Discussion. It would be valuable to discuss mechanisms, which can be responsible for variation of the disease severity in the affected dogs (incomplete penetrance or incomplete expressivity of the mutation, oligogenic inheritance, epigenetic mechanisms, etc?).
Response: We added a paragraph to the discussion, in which we stress once more that further studies are needed to clarify this important question. We speculate that the diet and possibly additional genetic factors are the most important modulators of disease severity, but this really needs to be investigated in future studies.